# Quantifying inequities in COVID-19 vaccine distribution over time by social vulnerability, race and ethnicity, and location: A population-level analysis in St. Louis and Kansas City, Missouri

**Aaloke Mody** [1] *, **Cory Bradley** [1], **Salil Redkar** [1], **Branson Fox** [1], **Ingrid Eshun-Wilson** [1], **Matifadza G. Hlatshwayo** [2], **Anne Trolard** [1,3], **Khai Hoan Tram** [4], **Lindsey M. Filiatreau** [1], **Franda Thomas** [2], **Matt Haslam** [2], **George Turabelidze** [5], **Vetta Sanders-Thompson** [6], **William G. Powderly** [1,3], **Elvin H. Geng** [1,3]

**1** Washington University School of Medicine, St. Louis, Missouri, United States of America, **2** St. Louis City Department of Health, St. Louis, Missouri, United States of America, **3** Institute for Public Health, Washington University in St. Louis, St. Louis, Missouri, United States of America, **4** University of Washington School of Medicine, Seattle, Washington, United States of America, **5** Missouri Department of Health and Senior Services, Jefferson City and St Louis, Missouri, United States of America, **6** Brown School of Social Work, Washington University in St. Louis, St. Louis, Missouri, United States of America

* aaloke.mody@wustl.edu

**Data Availability Statement:** The data underlying the results presented in the study are managed by

## Abstract

### Background

Equity in vaccination coverage is a cornerstone for a successful public health response to COVID-19. To deepen understanding of the extent to which vaccination coverage compares with initial strategies for equitable vaccination, we explore primary vaccine series and booster rollout over time and by race/ethnicity, social vulnerability, and geography.

### Methods and findings

We analyzed data from the Missouri Department of Health and Senior Services on all COVID-19 vaccinations administered across 7 counties in the St. Louis region and 4 counties in the Kansas City region. We compared rates of receiving the primary COVID-19 vaccine series and boosters relative to time, race/ethnicity, zip-code-level Social Vulnerability Index (SVI), vaccine location type, and COVID-19 disease burden. We adapted a well-established tool for measuring inequity—the Lorenz curve—to quantify inequities in COVID-19 vaccination relative to these key metrics. Between 15 December 2020 and 15 February 2022, 1,763,036 individuals completed the primary series and 872,324 received a booster. During early phases of the primary series rollout, Black and Hispanic individuals from high SVI zip codes were vaccinated at less than half the rate of White individuals from low SVI zip codes, but rates increased over time until they were higher than rates in White individuals after June 2021; Asian individuals maintained high levels of vaccination throughout.

the Missouri Department of Health and Senior Services and are not publicly available. Details on how to request data for public health research are available at https://health.mo.gov/data.

**Funding:** This work was supported by the National Center for Advancing Translational Sciences (KL2 TR002346 to AM and IEW) and National Institute of Allergy and Infectious Diseases (K24 AI134413 to EHG). This project has also been funded (in part) by the State of Missouri under a contract awarded to Washington University. The contents do not necessarily reflect the reviews and policies of the State of Missouri, nor does mention of trade names or commercial products constitute endorsement of recommendation for use. The funders had no role in study design, data collection and analysis, decision to publish, or preparation of the manuscript.

**Competing interests:** I have read the journal's policy and the authors of this manuscript have the following competing interests: EHG is a member of PLOS Medicine's Editorial Board. All other authors have declared no competing interests.

**Abbreviations:** aRR, adjusted rate ratio; SVI, Social Vulnerability Index.

Increasing vaccination rates in Black and Hispanic communities corresponded with periods when more vaccinations were offered at small community-based sites such as pharmacies rather than larger health systems and mass vaccination sites. Using Lorenz curves, zip codes in the quartile with the lowest rates of primary series completion accounted for 19.3%, 18.1%, 10.8%, and 8.8% of vaccinations while representing 25% of the total population, cases, deaths, or population-level SVI, respectively. When tracking Gini coefficients, these disparities were greatest earlier during rollout, but improvements were slow and modest and vaccine disparities remained across all metrics even after 1 year. Patterns of disparities for boosters were similar but often of much greater magnitude during rollout in fall 2021. Study limitations include inherent limitations in the vaccine registry dataset such as missing and misclassified race/ethnicity and zip code variables and potential changes in zip code population sizes since census enumeration.

## Conclusions

Inequities in the initial COVID-19 vaccination and booster rollout in 2 large US metropolitan areas were apparent across racial/ethnic communities, across levels of social vulnerability, over time, and across types of vaccination administration sites. Disparities in receipt of the primary vaccine series attenuated over time during a period in which sites of vaccination administration diversified, but were recapitulated during booster rollout. These findings highlight how public health strategies from the outset must directly target these deeply embedded structural and systemic determinants of disparities and track equity metrics over time to avoid perpetuating inequities in healthcare access.

## Author summary

### Why was this study done?

- Equitable vaccine strategies are critical for the public health response to COVID-19, but there is limited understanding of how vaccination campaigns compared to different metrics for equity.

- Many initial approaches to vaccine allocation sought to acknowledge known disparities in exposure risk, disease burden, needs, and access by formally considering social vulnerability or race/ethnicity in plans to prioritize vaccinations, but there is limited empirical evaluation of how actual primary vaccine series and subsequent booster efforts aligned with the initial goals set out for equity.

- We quantify COVID-19 vaccine-related inequities in receipt of the primary vaccine series and boosters across key equity metrics including race/ethnicity, social vulnerability, location, and time using a novel application of Lorenz curves and Gini coefficients —tools from economics to measure inequalities—in the St. Louis and Kansas City regions of Missouri.

### What did the researchers do and find?

- We analyzed data from the Missouri Department of Health and Senior Services on all COVID-19 vaccinations administered in the St. Louis and Kansas City regions. We compared rates of receiving the primary COVID-19 vaccine series and boosters relative to time, race/ethnicity, zip-code-level Social Vulnerability Index (SVI), vaccine location type, and COVID-19 disease burden. We adapted Lorenz curves and Gini coefficients to quantify the inequities in COVID-19 vaccination relative to these key metrics and examined how they changed over time.

- Black and Hispanic individuals from high SVI zip codes completed the primary series at less than half the rate of White individuals from low SVI zip codes during early phases of the primary series rollout, but their vaccination rates surpassed rates in White individuals after June 2021. These relative increases in primary series completion rates in Black and Hispanic communities corresponded to periods when vaccinations became more available at small community-based sites.

- Lorenz curves demonstrated that zip codes in the quartile with the lowest rates of primary series completion accounted for 19.3%, 18.1%, 10.8%, and 8.8% of vaccinations while representing 25% of the total population, cases, deaths, or population-level SVI, respectively. Tracking Gini coefficients over time demonstrated that these disparities were greatest earlier during rollout, but only improved slowly and modestly over time.

- Patterns of disparities for boosters were similar but often of much greater magnitude that those seen for completion of the primary vaccine series.

### What do these findings mean?

- Vaccination coverage for both the primary series and boosters demonstrated substantial disparities across race/ethnicity, levels of social vulnerability, and types of vaccine administration sites, and over time.

- Despite well-documented inequities for COVID-19 and the need for equitable vaccine approaches, the strategies employed did not overcome deeply entrenched systemic inequities in healthcare and society.

- Public health strategies must proactively target these deeply embedded structural determinants of disparities from the outset and should systematically track equity metrics over time to avoid perpetuating inequities in healthcare access.

## Introduction

The initial wave of coronavirus disease 2019 (COVID-19) redemonstrated and highlighted historical inequities in health by race and ethnicity and other social indicators of vulnerability [1–3], prompting a range of efforts to design public health services that redress inequity in the COVID-19 response. Across a wide range of indicators, disease burden as measured by COVID-19 cases, hospitalizations, and mortality has disproportionately affected minoritized

communities [1–3]. Initial responses to COVID-19 through established channels were thus accompanied by additional efforts to address the evolving disparities. Nevertheless, minoritized and vulnerable communities still had reduced access to testing and treatments and experienced disproportionate impacts of social distancing and lockdown policies on employment, education, and housing [4–8]. Against this backdrop, achieving equitable vaccination has been and continues to be one of the most critical public health challenges for mitigating the impact of the COVID-19 pandemic and achieving long-term control.

Closer examination of equity in the vaccine response evaluating the extent to which health systems performed in this domain is still necessary, and something that has not clearly been documented in the literature to date. Whereas equality simply refers to provision of equal resources to every individual regardless of need, equitable approaches acknowledge that individuals will have different risks, needs, or opportunities and that access to or distribution of resources needs to take these differences into account. Strategies and frameworks to guide the equitable allocation and distribution of vaccines were developed when vaccines for SARS-CoV-2 became available in December 2020 [9,10], but empirical examination of how actual primary vaccine series and subsequent booster efforts aligned with the initial goals set out for equity is still needed. For example, several strategies proposed formally considering geography, social vulnerability, or race/ethnicity in plans to prioritize and distribute vaccinations in response to the known inequities in exposure risk and disease burden across these metrics [11–13]. Examinations of equity must thus document patterns of vaccination across race/ethnicity, social vulnerability, and geography and over time, and how vaccinations are delivered, to understand the mechanisms that give rise to disparities and to yield key insights into successes, failures, and steps for redress to achieve equitable vaccination strategies.

In this paper, we deepen our understanding of COVID-19 vaccine-related disparities by examining inequities in vaccination in the St. Louis and Kansas City regions in Missouri—regions with a history of health disparities—across several key metrics. We characterize rates of receiving the primary vaccine series and boosters over time and by race/ethnicity, social vulnerability, disease burden, geography, and vaccination location type. We use Lorenz curves and Gini coefficients—tools from economics commonly used to measure inequity in a population—to quantify and track inequities in COVID-19 vaccination over time relative to different metrics for conceptualizing equity [14]. The novel application of this methodology—which we previously used to characterize COVID-19 testing disparities [4]—has potential to yield deeper insights into the progress made towards vaccine equity in these regions, which may then better inform health policy solutions to address remaining gaps.

## Methods

### Ethics statement

The study was approved by the institutional review board at Washington University in St. Louis (IRB ID# 202009021). The research in this paper was not prespecified and consists of secondary analysis of preexisting de-identified data. This paper was prepared according to STROBE guidelines (S1 STROBE Checklist).

### Study setting and data

We sought to assess disparities in COVID-19 vaccination across the 7 counties in the St. Louis region (St. Louis City, St. Louis County, St. Charles, Jefferson, Franklin, Lincoln, and Warren; total population 2,095,978: 19.2% Black, 73.1% White, 3.0% Hispanic, 3.2% Asian) and the 4 counties in the Kansas City region (Jackson, Clay, Cass, and Platte; total population 1,121,224: 16.8% Black, 73.2% White, 8.2% Hispanic, 2.0% Asian). These counties make up the broader

metropolitan areas located within Missouri for these 2 cities. Vaccines first became available on 15 December 2020, and all individuals became eligible on 29 March 2021. We used data from the Missouri Department of Health and Senior Services on SARS-CoV-2 vaccines administered in Missouri to individuals 12 years old and up between 15 December 2020 and 15 February 2022. Reporting of vaccinations was mandated, so this database is expected to contain near complete data on all vaccinations administered in Missouri. This individual-level dataset contains vaccination date, type, and dose number; administration site; and patient age, sex, race/ethnicity, and zip code, and was de-duplicated and cleaned by the Missouri Department of Health and Senior Services. We used 2020 census data to obtain age-, sex-, and race-stratified zip code population estimates and 2018 American Community Survey (ACS) data to obtain sociodemographic and socioeconomic characteristics of individual zip codes as well as the Centers for Disease Control and Prevention's Social Vulnerability Index (SVI). The SVI is a composite metric that captures a community's vulnerability to external stresses on human health and is calculated from 15 ACS variables measuring demographics, socioeconomic status, household composition, and infrastructure [15].

## Analyses

Our analyses seek to characterize patterns of disparities in receiving the primary vaccine series and boosters over time by examining rates of vaccination with respect to race/ethnicity and social vulnerability, changes in the types of locations vaccines were being administered, and the extent to which vaccine administration was equitable between zip codes. We adapted methods that we had previously used to assess disparities related to COVID-19 testing and extended them to COVID-19 vaccination [4].

First, we estimated the rates and cumulative incidence of COVID-19 vaccination over time stratifying individuals by race/ethnicity (i.e., Black, White, Hispanic, or Asian) and whether they lived in a zip code with a low, medium, or high SVI (i.e., less than 0.333, 0.333 to 0.666, or greater than or equal to 0.666, respectively). We examined completion rates for the primary vaccine series (defined as 2 doses of either BNT162b2 mRNA [Pfizer] or mRNA-1273 [Moderna] or a single dose of Ad26.COV2.S [Johnson & Johnson]) and boosters (defined as a single dose of any vaccine after completing the primary series).

Second, we examined vaccine distribution by the type of site at which individuals received their primary vaccine series and boosters over time and by race/ethnicity and zip-code-level SVI. We categorized vaccine administration sites into health facilities (e.g., clinics, hospitals, and health-system-affiliated sites) that administered a small, medium, or large volume of vaccinations (i.e., less than 1,000, 1,000 to 10,000, or greater than 10,000 unique individuals vaccinated, respectively), public health departments (including mass vaccination sites), pharmacies, employer/school-based sites, and other (e.g., dialysis centers, home health, nursing homes, mental health/psychiatric facilities, and correctional facilities).

Third, we generated modified versions of Lorenz curves to assess the relative equity in the distribution of COVID-19 vaccinations across zip codes. Lorenz curves—originally developed by economists to graphically represent income equality—have more recently been leveraged as a tool for public health [14,16,17]. Lorenz curves are generated by plotting the cumulative proportion of the total population against the cumulative proportion of a resource after sorting values in ascending order. The curve follows a straight line at a 45˚ angle when a resource is equitably distributed across the population and becomes more convex with increasing inequity. In general, equitable vaccination strategies seek to balance the number of vaccines with the overall risk of disease in a community, but the most appropriate metric of equity for so doing will depend on whether one considers the goal to be creating balance between

vaccination rates relative to the total population, overall disease burden (i.e., number of COVID-19 cases or deaths), or risk factors (i.e., social vulnerability) in a community. To examine vaccine equity from these different perspectives, we adapted the Lorenz curve method to examine disparities in receiving the primary vaccine series and boosters relative to several relevant metrics: (1) the total population, (2) the number of diagnosed COVID-19 cases, (3) the number of COVID-19 deaths, and (4) population-level social vulnerability, which we defined as the zip-code-level SVI multiplied by zip code population. For each curve, we calculated Gini coefficients—a measure of equality/inequality between 0 and 1, with 0 indicating perfect equality and 1 indicating perfect inequality—and assessed how these changed over time [18]. We also grouped zip codes into quartiles based on their position on Lorenz curves and assessed differences in zip-code-level sociodemographic and socioeconomic characteristics using Kruskal–Wallis tests.

Fourth, we generated bubble plots to compare primary vaccine series and booster completion rates for Black, Hispanic, and Asian residents relative to White residents living in the same zip code. For these analyses, we considered only zip codes whose populations had at least 25 individuals from each of the racial/ethnic groups to avoid extreme outliers from small denominators.

Lastly, we performed univariate and multivariable mixed-effects Poisson regression to identify individual (e.g., sex, race/ethnicity, age) and zip-code-level (e.g., SVI, racial makeup, health insurance coverage) factors independently associated with receiving the primary vaccine series and boosters; in multivariable models, we excluded zip-code-level variables that would be expected to relate directly to SVI (e.g., poverty and median income). We applied an established method for using Poisson regression with robust variances to estimate risk ratios from binary outcomes [19,20]. We leveraged vaccination and 2020 census data to estimate the number of unvaccinated individuals across strata of age, sex, and race/ethnicity in each zip code. We visually assessed for linearity in the relationship between continuous variables and outcomes and present variables with nonlinear relationships as categorical variables (i.e., age and zip code SVI). The effect of race/ethnicity and racism on health outcomes is mediated by (as opposed to confounded by) ecological structural factors such socioeconomic status; thus, unadjusted analyses assess the overall association with race/ethnicity and racism while adjusted analyses can be thought of as assessing the contribution of systemic racism that still remains after adjusting for the mediating effects of measured ecological factors [21–23].

To account for missingness in race/ethnicity and patient zip code variables, we performed multiple imputation using multivariate normal imputation methods ($n = 50$ imputations) [24–26]. For zip codes, we first transformed them to the latitude and longitude of their centroid, ran the multiple imputation model, and then transformed multiply imputed latitude and longitude values back into zip codes. Missingness was highly dependent on vaccination date and administration site, and thus the missing at random assumption required for unbiased imputation (i.e., that missingness was random conditional on all the variables included in the imputation model [administration site, vaccination date, sex, age, race/ethnicity, zip code latitude and longitude, type of vaccine]) was very plausible in our setting [24–26].

All analyses were conducted using Stata/MP 17.0 and R 3.2.4.

## Results

Between 15 December 2020 to 15 February 2022, 4,741,806 total COVID-19 vaccines were administered to 2,019,715 unique individuals across the 7 counties in the St. Louis region and the 4 counties in the Kansas City region. Among those receiving at least 1 dose in St. Louis and Kansas City, 1,763,036 (87.3%) completed the primary series, and 872,324 (43.2%) received a

booster. Of those who completed the primary series, 81.2% of individuals did so by 15 June 2021 (Tables 1, 2, S1, and S2).

## Rates of COVID-19 primary and booster vaccination by race/ethnicity and SVI over time

The rate of primary COVID-19 vaccination steadily increased until peaking in mid-April 2021. This was followed by rapid decline, with smaller upticks at the end of May 2021 and then during the Delta wave beginning in July 2021; there was no corresponding uptick in vaccination rates during the Omicron wave beginning in mid-December 2021 (Figs 1 and S1–S4; S3 Table). Up through April 2021, White individuals from zip codes with low SVI were vaccinated at a rate greater than 2 times that of Black and Hispanic individuals from high SVI zip codes, but the rate ratio declined over time. Asian individuals from all zip codes were vaccinated at the highest rates. During the same early period, Black and Hispanic individuals from low SVI zip codes were vaccinated at rates somewhat similar to or higher than those of White individuals from medium and high SVI zip codes. After June 2021, Black and Hispanic individuals from high, medium, and low SVI zip codes were vaccinated at higher rates than White individuals, although this was also during periods with lower absolute numbers of vaccinations (Figs 1 and S1–S4; S3 Table). Patterns were largely similar across St. Louis and Kansas City (S2 and S3 Figs).

Booster rates increased starting in October 2021 and peaked in early December 2021 at the beginning of the Omicron wave, albeit at much lower levels than for the primary vaccine series, and started to decline in January 2022. Patterns of disparities across race/ethnicity were similar for boosters and completion of the primary series (Figs 1 and S1–S4; S3 Table).

## Locations of COVID-19 vaccination over time

Early during the vaccination campaign, the vast majority of vaccines were delivered through medium and large volume health facilities (Fig 2). From February through April 2021, a substantial proportion were also delivered through public health departments (including mass vaccination sites). After April 2021, the proportion of vaccines administered through pharmacies steadily increased, accounting for about 70% of vaccines administered after July 2021. Black individuals received comparatively more vaccines through employer/school-sponsored sites, small volume health facilities, or other facilities such as dialysis centers, home health, and nursing homes, and fewer from pharmacies and health departments. Hispanic and Asian individuals received comparatively more vaccines through pharmacies and health departments; Hispanic individuals also received relatively few vaccines from large volume health facilities. Again, patterns were qualitatively similar for boosters (Fig 2).

## COVID-19 vaccine disparities across zip codes using Lorenz curves

Modified Lorenz curves depict the distribution of COVID-19 vaccinations with respect to the total population, diagnosed COVID-19 cases, COVID-19 deaths, and population-level SVI across zip codes (Fig 3). For the primary vaccine series, zip codes in the quartile with the lowest rates of vaccinations accounted for 19.3%, 18.1%, 10.8%, and 8.8% of vaccines while representing 25% of the total population, cases, deaths, or population-level SVI, respectively. These zip codes, in general, had higher proportions of Black residents, lower median incomes, higher rates of poverty, lower rates of health insurance coverage, a higher proportion of residents employed in the service sector, and a higher rate of COVID-19 deaths (Fig 3; S4–S7 Tables). In contrast, zip codes with the highest rates of vaccination accounted for 30.7%, 35.0%, 44.2%, and 56.1% of vaccinations while representing 25% of the total population, cases, deaths, or

**Table 1. Characteristics of individuals completing the primary series.**

| Characteristic | Overall (n = 1,763,036) | Black (n = 226,520) | White (n = 1,089,138) | Hispanic (n = 60,079) | Asian (n = 47,842) | High SVI (n = 202,822) | Medium SVI (n = 527,677) | Low SVI (n = 1,032,218) |
|---|---|---|---|---|---|---|---|---|
| **Sex**\*, **n (%)** | | | | | | | | |
| Male | 797,909 (45.3%) | 92,565 (40.9%) | 503,427 (46.2%) | 29,867 (49.7%) | 22,418 (46.9%) | 86,641 (42.8%) | 237,048 (45.0%) | 474,051 (46.0%) |
| Female | 963,261 (54.7%) | 133,897 (59.1%) | 585,515 (53.8%) | 30,179 (50.3%) | 25,400 (53.1%) | 115,946 (57.2%) | 289,832 (55.0%) | 557,335 (54.0%) |
| **Age category**\*, **n (%)** | | | | | | | | |
| 12–19 years | 160,216 (9.1%) | 23,041 (10.2%) | 93,963 (8.6%) | 8,365 (13.9%) | 5,227 (10.9%) | 18,727 (9.2%) | 42,966 (8.1%) | 98,503 (9.5%) |
| 20–34 years | 331,936 (18.8%) | 41,758 (18.4%) | 199,525 (18.3%) | 16,722 (27.8%) | 14,901 (31.1%) | 39,559 (19.5%) | 105,291 (20.0%) | 186,960 (18.1%) |
| 35–44 years | 257,408 (14.6%) | 32,393 (14.3%) | 156,842 (14.4%) | 11,324 (18.8%) | 9,133 (19.1%) | 29,400 (14.5%) | 74,321 (14.1%) | 153,638 (14.9%) |
| 45–54 years | 251,004 (14.2%) | 36,568 (16.1%) | 151,349 (13.9%) | 9,403 (15.6%) | 7,800 (16.3%) | 29,668 (14.6%) | 73,131 (13.9%) | 148,161 (14.4%) |
| 55–64 years | 304,054 (17.2%) | 43,336 (19.1%) | 192,128 (17.6%) | 7,432 (12.4%) | 5,185 (10.8%) | 37,112 (18.3%) | 93,520 (17.7%) | 173,368 (16.8%) |
| 65–74 years | 263,353 (14.9%) | 31,420 (13.9%) | 170,469 (15.7%) | 4,342 (7.2%) | 3,502 (7.3%) | 29,352 (14.5%) | 80,562 (15.3%) | 153,402 (14.9%) |
| 75+ years | 195,065 (11.1%) | 18,004 (7.9%) | 124,862 (11.5%) | 2,486 (4.1%) | 2,094 (4.4%) | 18,964 (9.4%) | 57,886 (11.0%) | 118,186 (11.4%) |
| **Race/ethnicity**\*, **n (%)** | | | | | | | | |
| Black | 226,520 (13.3%) | — | — | — | — | 99,431 (50.7%) | 84,173 (16.6%) | 42,860 (4.3%) |
| White | 1,089,138 (64.1%) | — | — | — | — | 48,758 (24.8%) | 310,296 (61.4%) | 729,935 (73.2%) |
| Hispanic | 60,079 (3.5%) | — | — | — | — | 13,045 (6.7%) | 19,738 (3.9%) | 27,288 (2.7%) |
| Asian | 47,842 (2.8%) | — | — | — | — | 3,319 (1.7%) | 13,333 (2.6%) | 31,174 (3.1%) |
| Other | 275,054 (16.2%) | — | — | — | — | 31,689 (16.2%) | 78,010 (15.4%) | 165,302 (16.6%) |
| **Median zip code SVI (IQR)** | 0.29 (0.16, 0.47) | 0.57 (0.41, 0.79) | 0.25 (0.15, 0.42) | 0.37 (0.20, 0.63) | 0.24 (0.15, 0.44) | 0.79 (0.75, 0.86) | 0.47 (0.41, 0.53) | 0.18 (0.13, 0.25) |
| **Vaccine location type, n (%)** | | | | | | | | |
| Small volume health facility | 53,828 (3.1%) | 9,477 (4.2%) | 31,848 (2.9%) | 1,632 (2.7%) | 983 (2.1%) | 7,638 (3.8%) | 16,854 (3.2%) | 29,315 (2.8%) |
| Medium volume health facility | 246,975 (14.0%) | 28,443 (12.6%) | 156,876 (14.4%) | 8,632 (14.4%) | 5,746 (12.0%) | 25,482 (12.6%) | 72,552 (13.7%) | 148,901 (14.4%) |
| Large volume health facility | 423,980 (24.0%) | 51,549 (22.8%) | 276,770 (25.4%) | 7,798 (13.0%) | 10,019 (20.9%) | 39,691 (19.6%) | 115,636 (21.9%) | 268,614 (26.0%) |
| Pharmacy | 655,285 (37.2%) | 80,441 (35.5%) | 390,072 (35.8%) | 28,621 (47.6%) | 18,317 (38.3%) | 81,837 (40.3%) | 205,835 (39.0%) | 367,514 (35.6%) |
| Health department | 304,999 (17.3%) | 40,798 (18.0%) | 192,537 (17.7%) | 11,171 (18.6%) | 10,606 (22.2%) | 36,329 (17.9%) | 93,509 (17.7%) | 175,068 (17.0%) |
| Employer/school | 39,286 (2.2%) | 7,960 (3.5%) | 21,035 (1.9%) | 981 (1.6%) | 1,614 (3.4%) | 5,701 (2.8%) | 12,397 (2.3%) | 21,175 (2.1%) |
| Other | 38,683 (2.2%) | 7,852 (3.5%) | 20,000 (1.8%) | 1,244 (2.1%) | 557 (1.2%) | 6,144 (3.0%) | 10,894 (2.1%) | 21,631 (2.1%) |
| **Primary series vaccine type, n (%)** | | | | | | | | |
| J&J | 115,409 (6.5%) | 18,363 (8.1%) | 71,603 (6.6%) | 4,658 (7.8%) | 2,271 (4.7%) | 16,193 (8.0%) | 37,860 (7.2%) | 61,316 (5.9%) |
| Moderna | 485,296 (27.5%) | 59,402 (26.2%) | 286,140 (26.3%) | 16,355 (27.2%) | 10,988 (23.0%) | 59,493 (29.3%) | 156,016 (29.6%) | 269,695 (26.1%) |
| Pfizer | 1,162,331 (65.9%) | 148,755 (65.7%) | 731,395 (67.2%) | 39,066 (65.0%) | 34,583 (72.3%) | 127,136 (62.7%) | 333,801 (63.3%) | 701,207 (67.9%) |
| **Booster received, n (%)** | 872,324 (49.5%) | 84,564 (37.3%) | 569,411 (52.3%) | 20,094 (33.4%) | 23,411 (48.9%) | 74,292 (36.6%) | 245,129 (46.5%) | 552,779 (53.6%) |

*(Continued)*

**Table 1.** (Continued)

| Characteristic | Overall (*n* = 1,763,036) | Black (*n* = 226,520) | White (*n* = 1,089,138) | Hispanic (*n* = 60,079) | Asian (*n* = 47,842) | High SVI (*n* = 202,822) | Medium SVI (*n* = 527,677) | Low SVI (*n* = 1,032,218) |
|---|---|---|---|---|---|---|---|---|
| Time period, *n* (%) | | | | | | | | |
| 15 Dec 2020–15 Jun 2021 | 1,431,263 (81.2%) | 153,724 (67.9%) | 916,653 (84.2%) | 43,003 (71.6%) | 41,484 (86.7%) | 138,841 (68.5%) | 415,767 (78.8%) | 876,398 (84.9%) |
| 16 Jun 2021–15 Dec 2021 | 311,744 (17.7%) | 67,342 (29.7%) | 162,831 (14.9%) | 15,699 (26.1%) | 5,815 (12.2%) | 59,482 (29.3%) | 105,018 (19.9%) | 147,189 (14.3%) |
| 16 Dec 2021–15 Feb 2022 | 20,029 (1.1%) | 5,454 (2.4%) | 9,654 (0.9%) | 1,377 (2.3%) | 543 (1.1%) | 4,499 (2.2%) | 6,892 (1.3%) | 8,631 (0.8%) |

*Overall missing values: sex, 1,866; race, 64,403; zip code, 319.

J&J, Johnson & Johnson; SVI, Social Vulnerability Index.

population-level SVI, respectively. These zip codes tended to have a lower percentage of Black residents and to be more socioeconomically advantaged (Fig 3; S4–S7 Tables). These patterns were similar, but demonstrated a greater magnitude of disparities, for boosters (Fig 3; S4–S7 Tables).

When examining changes in Gini coefficients and vaccine inequities between zip codes over time, inequities were extremely high during the initial periods of the primary series roll-out, but began to slowly decrease after February 2021 relative to population, deaths, and total social vulnerability, but improvements relative to diagnosed cases plateaued around May 2021. Nevertheless, these improvements were slow and modest, and vaccine inequities between zip codes remained substantial for all metrics through January 2022 (Figs 4 and S5). With respect to boosters, Gini coefficients once again were very high in the beginning of rollout, followed by slow improvement relative to population, cases, and deaths; Gini coefficients did not improve (and even worsened initially) relative to total social vulnerability (Figs 4 and S6). There were limited improvements after December 2021 during the Omicron wave.

## COVID-19 vaccine disparities within zip codes

In zip codes with lower vaccination coverage (which also tended to have higher SVI), Black, Hispanic, and Asian individuals generally had lower rates of primary series completion than White individuals residing in the same zip code (Figs 5 and S7). However, in zip codes with high vaccine coverage (which also tended to have low SVI), Black, Hispanic, and Asian individuals often had higher primary series completion than White individuals in the same zip code. For boosters, Black and Hispanic individuals had lower vaccination rates than White individuals across most zip codes, although Asian individuals tended slightly to have higher booster rates (Figs 5 and S7).

## Factors associated with receiving the primary vaccine series and boosters

In multivariable mixed-effects Poisson regression, Black and Hispanic individuals had slightly lower rates of completing the primary vaccine series compared to White individuals (adjusted rate ratio [aRR] 0.94 [95% CI 0.93–0.94] and 0.96 [95% CI 0.95–0.97], respectively), while Asian individuals had slightly higher rates (aRR 1.03 [95% CI 1.02–1.03]). Living in a medium or high SVI zip code was also associated with a lower vaccination rate compared to living in a low SVI zip code (aRR 0.92 [95% CI 0.91–0.92] and 0.88 [95% CI 0.88–0.89], respectively) (Table 3). Additional factors associated with increased vaccination were being female and being 12 to 19 years old or 55 years old or older (as compared to 45 to 54 years old); individuals

**Table 2. Characteristics of individuals receiving a booster vaccination.**

| Characteristic | Overall (n = 872,324) | Black (n = 84,564) | White (n = 569,411) | Hispanic (n = 20,094) | Asian (n = 23,411) | High SVI (n = 74,292) | Medium SVI (n = 245,129) | Low SVI (n = 552,779) |
|---|---|---|---|---|---|---|---|---|
| **Sex*, n (%)** | | | | | | | | |
| Male | 375,780 (43.1%) | 32,955 (39.0%) | 250,231 (43.9%) | 9,324 (46.4%) | 10,758 (46.0%) | 29,909 (40.3%) | 104,287 (42.6%) | 241,523 (43.7%) |
| Female | 496,405 (56.9%) | 51,605 (61.0%) | 319,154 (56.1%) | 10,769 (53.6%) | 12,647 (54.0%) | 44,371 (59.7%) | 140,778 (57.4%) | 311,193 (56.3%) |
| **Age category*, n (%)** | | | | | | | | |
| 12–19 years | 41,139 (4.7%) | 3,290 (3.9%) | 26,448 (4.6%) | 1,589 (7.9%) | 1,826 (7.8%) | 2,372 (3.2%) | 8,904 (3.6%) | 29,858 (5.4%) |
| 20–34 years | 110,416 (12.7%) | 7,706 (9.1%) | 71,982 (12.6%) | 3,986 (19.8%) | 6,181 (26.4%) | 7,926 (10.7%) | 32,664 (13.3%) | 69,803 (12.6%) |
| 35–44 years | 111,695 (12.8%) | 8,628 (10.2%) | 73,872 (13.0%) | 3,443 (17.1%) | 4,616 (19.7%) | 7,855 (10.6%) | 28,751 (11.7%) | 75,067 (13.6%) |
| 45–54 years | 118,579 (13.6%) | 13,504 (16.0%) | 75,541 (13.3%) | 3,542 (17.6%) | 4,299 (18.4%) | 10,062 (13.5%) | 31,484 (12.8%) | 77,018 (13.9%) |
| 55–64 years | 169,867 (19.5%) | 20,789 (24.6%) | 110,789 (19.5%) | 3,502 (17.4%) | 2,927 (12.5%) | 16,817 (22.6%) | 49,656 (20.3%) | 103,369 (18.7%) |
| 65–74 years | 181,986 (20.9%) | 19,328 (22.9%) | 120,069 (21.1%) | 2,534 (12.6%) | 2,202 (9.4%) | 17,545 (23.6%) | 53,781 (21.9%) | 110,642 (20.0%) |
| 75+ years | 138,642 (15.9%) | 11,319 (13.4%) | 90,710 (15.9%) | 1,498 (7.5%) | 1,360 (5.8%) | 11,715 (15.8%) | 39,889 (16.3%) | 87,022 (15.7%) |
| **Race/ethnicity*, n (%)** | | | | | | | | |
| Black | 84,564 (9.9%) | — | — | — | — | 33,706 (46.2%) | 32,536 (13.6%) | 18,306 (3.4%) |
| White | 56,9411 (66.5%) | — | — | — | — | 21,793 (29.9%) | 152,881 (63.8%) | 394,667 (72.6%) |
| Hispanic | 20,094 (2.3%) | — | — | — | — | 2,739 (3.8%) | 6,190 (2.6%) | 11,161 (2.1%) |
| Asian | 23,411 (2.7%) | — | — | — | — | 1,255 (1.7%) | 6,166 (2.6%) | 15,986 (2.9%) |
| Other | 159,118 (18.6%) | — | — | — | — | 13,505 (18.5%) | 41,824 (17.5%) | 103,764 (19.1%) |
| **Median zip code SVI (IQR)** | 0.25 (0.16, 0.47) | 0.57 (0.34, 0.77) | 0.23 (0.15, 0.38) | 0.31 (0.16, 0.48) | 0.22 (0.15, 0.42) | 0.79 (0.71, 0.86) | 0.47 (0.41, 0.53) | 0.17 (0.12, 0.24) |
| **Booster location type, n (%)** | | | | | | | | |
| Small volume health facility | 41,300 (4.7%) | 7,073 (8.4%) | 25,989 (4.6%) | 808 (4.0%) | 954 (4.1%) | 5,210 (7.0%) | 11,627 (4.7%) | 24,456 (4.4%) |
| Medium volume health facility | 86,703 (9.9%) | 12,227 (14.5%) | 51,780 (9.1%) | 1,815 (9.0%) | 2,289 (9.8%) | 9,607 (12.9%) | 21,987 (9.0%) | 55,100 (10.0%) |
| Large volume health facility | 71,385 (8.2%) | 9,290 (11.0%) | 47,164 (8.3%) | 956 (4.8%) | 1,856 (7.9%) | 6,614 (8.9%) | 19,927 (8.1%) | 44,839 (8.1%) |
| Pharmacy | 610,285 (70.0%) | 46,383 (54.8%) | 409,591 (71.9%) | 14,519 (72.3%) | 17,050 (72.8%) | 44,096 (59.4%) | 169,980 (69.3%) | 396,131 (71.7%) |
| Health department | 37,460 (4.3%) | 6,514 (7.7%) | 21,900 (3.8%) | 1,369 (6.8%) | 607 (2.6%) | 6,001 (8.1%) | 14,181 (5.8%) | 17,260 (3.1%) |
| Employer/school | 6,469 (0.7%) | 591 (0.7%) | 3,767 (0.7%) | 177 (0.9%) | 473 (2.0%) | 383 (0.5%) | 2,108 (0.9%) | 3,976 (0.7%) |
| Other | 18,722 (2.1%) | 2,486 (2.9%) | 9,220 (1.6%) | 450 (2.2%) | 182 (0.8%) | 2,381 (3.2%) | 5,319 (2.2%) | 11,017 (2.0%) |
| **Booster vaccine type, n (%)** | | | | | | | | |
| J&J | 8,801 (1.0%) | 1,748 (2.1%) | 5,188 (0.9%) | 306 (1.5%) | 118 (0.5%) | 1,365 (1.8%) | 2,972 (1.2%) | 4,460 (0.8%) |
| Moderna | 293,809 (33.7%) | 26,933 (31.8%) | 189,263 (33.2%) | 7,175 (35.7%) | 7,127 (30.4%) | 26,406 (35.5%) | 86,414 (35.3%) | 180,946 (32.7%) |
| Pfizer | 569,714 (65.3%) | 55,883 (66.1%) | 374,960 (65.9%) | 12,613 (62.8%) | 16,166 (69.1%) | 46,521 (62.6%) | 155,743 (63.5%) | 367,373 (66.5%) |
| **Booster time period, n (%)** | | | | | | | | |
| 15 Dec 2020–15 Jun 2021 | 9,722 (1.1%) | 1,079 (1.3%) | 6,102 (1.1%) | 260 (1.3%) | 206 (0.9%) | 950 (1.3%) | 2,749 (1.1%) | 6,019 (1.1%) |

*(Continued)*

**Table 2.** (Continued)

| Characteristic | Overall (n = 872,324) | Black (n = 84,564) | White (n = 569,411) | Hispanic (n = 20,094) | Asian (n = 23,411) | High SVI (n = 74,292) | Medium SVI (n = 245,129) | Low SVI (n = 552,779) |
|---|---|---|---|---|---|---|---|---|
| 16 Jun 2021–15 Dec 2021 | 592,768 (68.0%) | 50,098 (59.2%) | 390,295 (68.5%) | 11,318 (56.3%) | 13,512 (57.7%) | 45,797 (61.6%) | 164,223 (67.0%) | 382,674 (69.2%) |
| 16 Dec 2021–15 Feb 2022 | 269,834 (30.9%) | 33,387 (39.5%) | 173,014 (30.4%) | 8,516 (42.4%) | 9,693 (41.4%) | 27,545 (37.1%) | 78,157 (31.9%) | 164,086 (29.7%) |

*Overall missing values: sex, 139; race/ethnicity, 15,726; zip code, 124.

J&J, Johnson & Johnson; SVI, Social Vulnerability Index.

20 to 34 years old had decreased vaccination rates. Differences in receipt of a booster vaccine were substantially higher across race, age, sex, and zip code SVI compared to the differences in completion of the primary vaccine series, except that 12- to 19-year-olds were less likely to receive a booster (Table 3).

## Discussion

Our analyses characterized disparities in the COVID-19 vaccination campaign in the St. Louis and Kansas City regions across racial/ethnic communities, across levels of social vulnerability, over time, and across types of vaccine administration sites. We describe changes in the rates of receiving the primary COVID-19 vaccination series and boosters across race/ethnicity and social vulnerability and highlight how these changes corresponded with shifts in the types of locations where individuals were vaccinated. We also use Lorenz curves and Gini coefficients to quantify disparities in vaccinations with respect to population, COVID-19-related disease burden, and social vulnerability. Overall, these results provide a deeper characterization the systemic inequities in distribution of one of the most critical (and initially scarce) resources for controlling the COVID-19 pandemic but one that is immediately actionable: COVID-19 vaccinations.

These analyses provide a deeper understanding of the patterns of vaccine inequities over time, and we note that disparities were greatest earlier on but have also largely persisted over time, with minimal improvement since April 2020. Furthermore, they emerged anew with the booster rollout in fall 2021. Early during vaccination, rates of completing the primary vaccine series were highest among White and Asian individuals in zip codes with low SVI. During this early period, a vast majority of vaccines were administered through health systems and also mass vaccination sites coordinated by public health departments. The relationship between race/ethnicity and zip code SVI is salient during this period: Black and Hispanic individuals living in high SVI zip codes had strikingly lower rates of vaccination compared to other groups, whereas Black and Hispanic individuals in low SVI zip codes had similar to somewhat higher rates of vaccination compared to White individuals in medium and high SVI zip codes. Over time, and particularly after all adults became eligible for vaccination, rates of vaccination among Black and Hispanic individuals across all SVI zip codes started to exceed those among White individuals. During these periods, sites of vaccine administration also diversified and shifted more towards pharmacies and other small community-based sites (and were much less likely to be at very large facilities). When quantifying these disparities using Lorenz curves, we note that disparities in vaccinations were highest relative to population-level social vulnerability and deaths, but still evident—albeit reduced—even when considering vaccinations relative to the overall population and diagnosed COVID-19 cases. Lastly, when examining disparities within zip codes, we see consistently higher rates of vaccination among White individuals

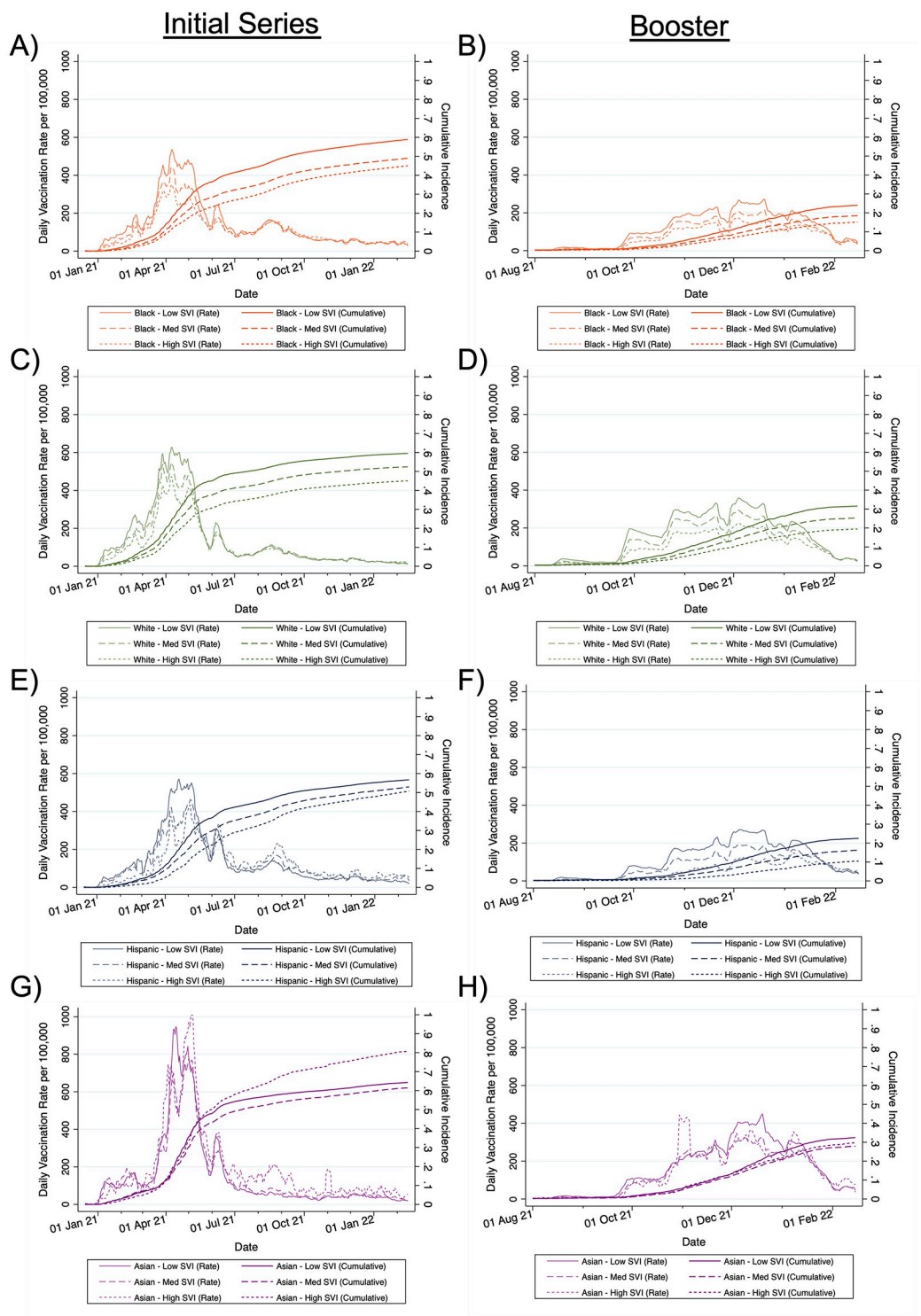

**Fig 1. Rates and cumulative incidence of receiving the primary COVID-19 vaccination series and boosters by race/ethnicity and SVI over time.** Initial series and booster vaccinations for Black (A and B), White (C and D), Hispanic (E and F), and Asian (G and H) individuals. Estimates represent 7-day moving averages derived from multiply imputed datasets. Denominators represent the total population aged 12 years and older. Low SVI indicates zip codes with SVI less than 0.333, medium SVI indicates zip codes with SVI between 0.333 and 0.666, and high SVI indicates zip codes with SVI greater than 0.666. SVI, Social Vulnerability Index.

compared to Black individuals, with the starkest difference in high SVI zip codes. Unfortunately, despite the slow progress from the early periods in improving equity in completion of the primary vaccine series, the same patterns of disparities were repeated again during the booster rollout, and were often of greater magnitude.

It is critical to understand these trends in the context of the underlying structural driving forces and decisions leading to these vaccination patterns, both of which are relevant nationally and not specific to Missouri. First, the high levels of disparities seen in the earlier stages of the primary vaccine series and booster rollouts likely reflect the fact that healthcare workers and older individuals were eligible for vaccination first, factors that are also associated with higher socioeconomic status and lower SVI [9,10]. Second, the early phases of the primary vaccine series rollout occurred primarily at sites associated with large health systems. However, these are also the sites at which Black and Hispanic individuals—and particularly those from high SVI zip codes—were comparatively less likely to ultimately receive vaccinations, highlighting a critical issue related to vaccine access among racially and ethnically marginalized and socially vulnerable communities [27–32]. Although large health systems may have been more readily able to overcome logistic issues and provide the robust cold chain needed for mRNA vaccines, they have limited mandates and expertise for implementing large-scale public health initiatives. Even prior to the pandemic, the significant disparities in who accesses care at these health systems and who is outside of them were well-known [30,33]. Physical access, challenges with scheduling (particularly online), disparities in insurance, lack of community partnerships, and mistrust of large institutions that have largely neglected underserved

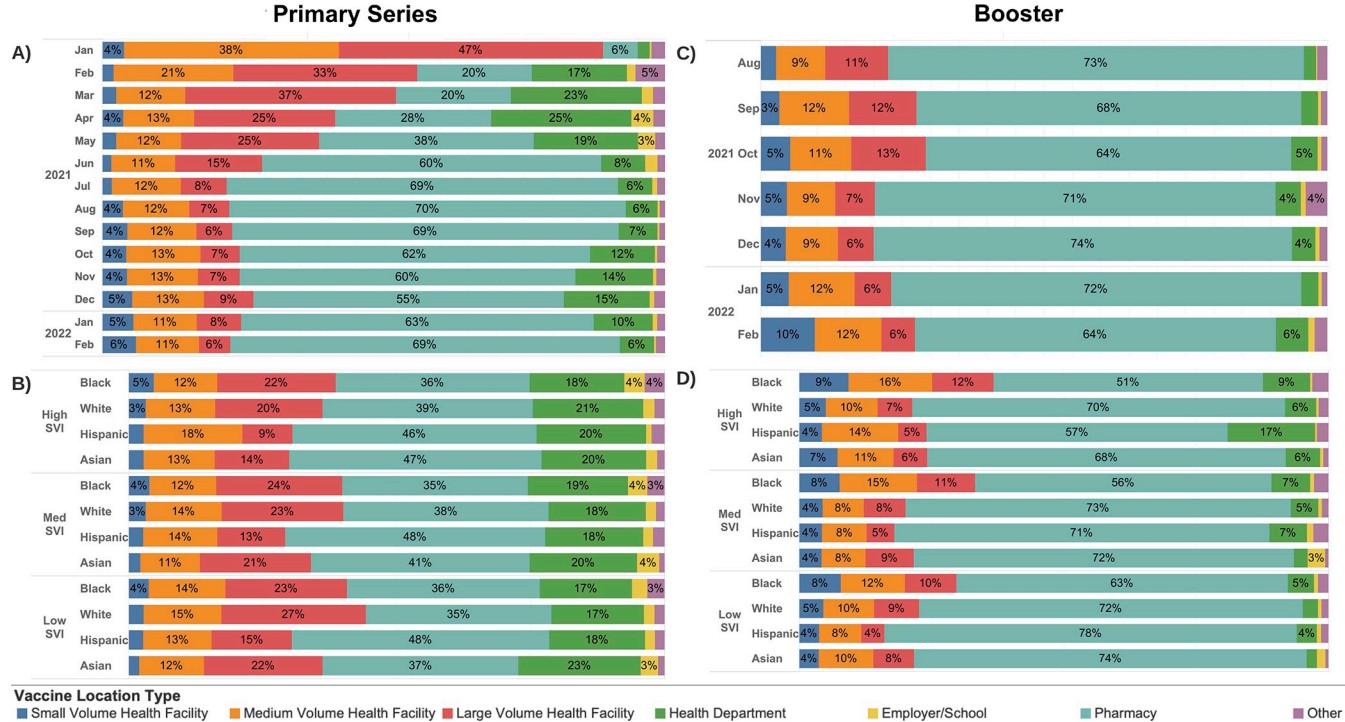

**Fig 2. Distribution of primary COVID-19 vaccine series and boosters by location type over time and by SVI and race/ethnicity.** Primary series and booster vaccination over time (A and C) and by SVI and race/ethnicity (B and D). Low SVI indicates zip codes with SVI less than 0.333, medium SVI indicates zip codes with SVI between 0.333 and 0.666, and high SVI indicates zip codes with SVI greater than 0.666. Health facilities were categorized as small, medium, or large volume based on whether they vaccinated less than 1,000, 1,000 to 10,000, or greater than 10,000 unique individuals. Other facilities included dialysis centers, home health, nursing homes, mental health/psychiatric facilities, and correctional facilities. Primary series vaccines were allocated to the location where the series was completed. SVI, Social Vulnerability Index.

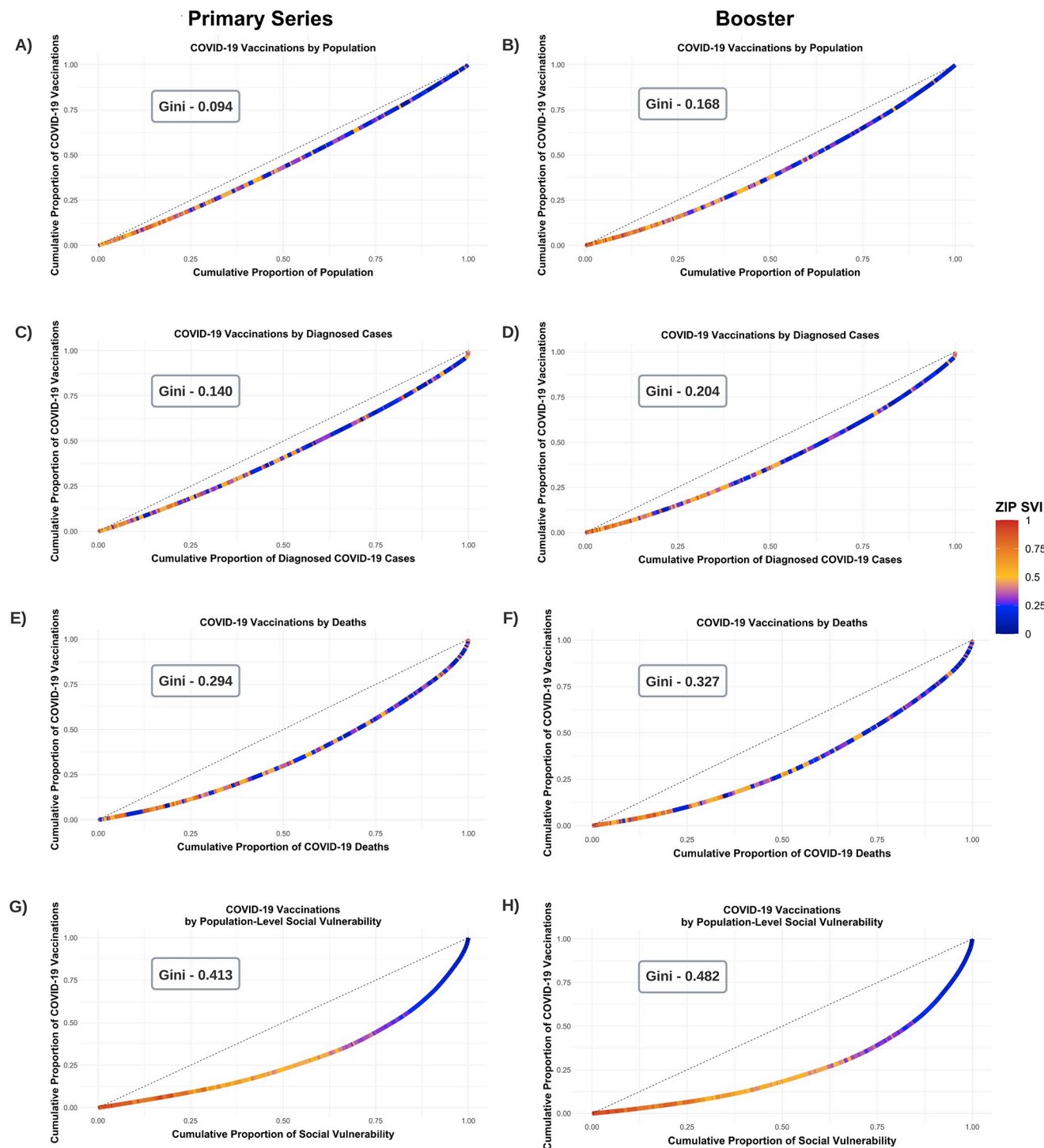

**Fig 3. Lorenz curves of disparities in COVID-19 vaccinations.** This figure depicts modified Lorenz curves examining disparities in COVID-19 vaccinations as of 15 February 2022. The units of analysis are zip codes, and they are color-coded by their SVI. The dashed line represents equitable distribution, where 50% of vaccinations are distributed in zip codes accounting for 50% of the population, cases, deaths, or total social vulnerability. The Lorenz curves measure disparities in the distribution of receiving the primary vaccine series or a booster relative to the total population aged 12 years or older (A and B), diagnosed COVID-19 cases (C and D), COVID-19 deaths (E and F), and total social vulnerability (G and H). SVI, Social Vulnerability Index.

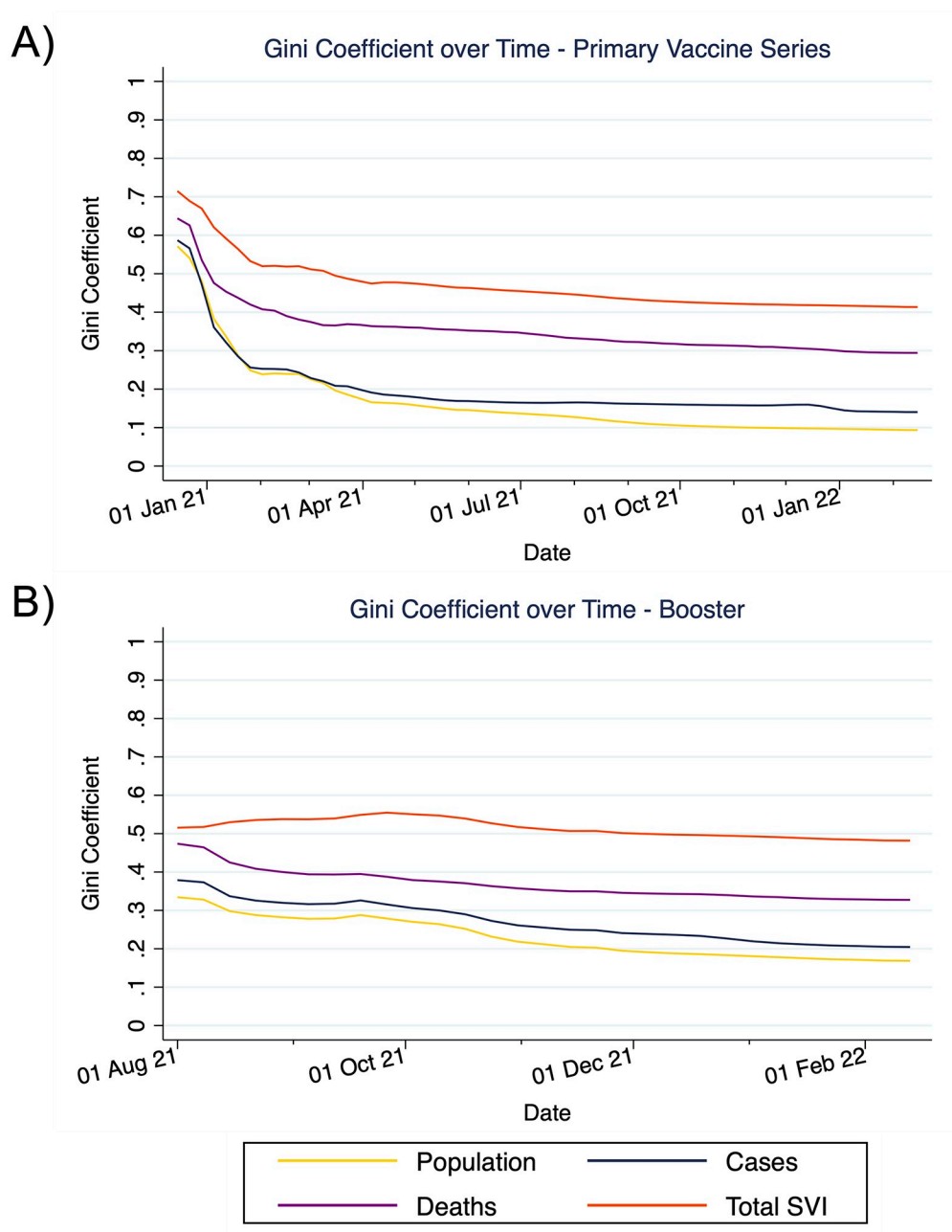

**Fig 4. Temporal patterns in COVID-19 vaccine inequities.** This figure depicts patterns in the Gini coefficients over time for inequities in receiving the primary vaccine series (A) and a booster (B) relative to population, diagnosed COVID-19 cases, COVID-19 deaths, and population-level social vulnerability. Gini coefficients were calculated on a weekly basis from Lorenz curves generated up through that time interval. SVI, Social Vulnerability Index.

communities often serve as salient barriers to care-seeking in large health systems for individuals from high SVI communities [27,30,34,35]. Vaccination campaigns are a public health strategy that requires broad reach into communities that large health systems do not have and were not designed for; thus, the strategies relying on these systems did not reach the most vulnerable

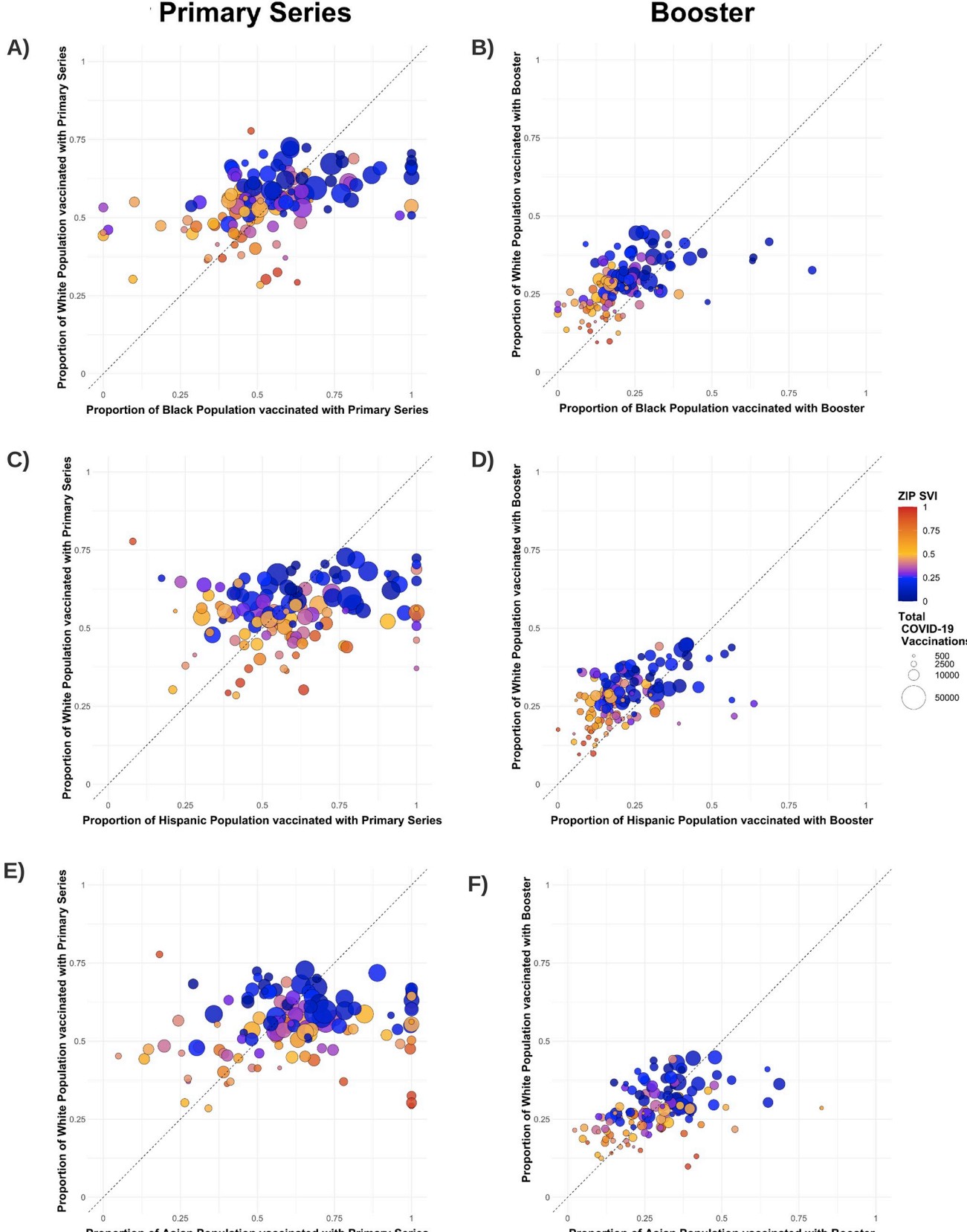

**Fig 5. Disparities in COVID-19 primary vaccine series and boosters among Black, Hispanic, and Asian versus White residents of the same zip code.**
This figure depicts vaccination rates for the primary series and boosters for Black (A and B), Hispanic (C and D), and Asian (E and F) residents compared to the White residents of the same zip code. Each marker represents a single zip code. Markers are color-coded by the zip code SVI and sized by the total number of vaccines administered in the zip code. The dashed line represents equitable vaccine distribution between the racial/ethnic groups being compared. Points above the dashed line indicate that there was decreased vaccination in Black, Hispanic, or Asian residents compared to White residents (and vice versa). SVI, Social Vulnerability Index.

populations essentially by design, even though the vaccines themselves were freely available. These patterns seen in both the primary vaccine series and booster rollout were also mirrored in prior research from our group examining disparities in COVID-19 testing, and their origins can be traced back to many of the same root causes [4]. Ultimately, the repeated reliance on systems with a history of providing lower access to certain segments of the population is representative of how structural inequities also became embedded in COVID-19 vaccine rollout from its onset and serves as a cautionary tale, albeit one that has been told too many times before.

Overall vaccination rates and patterns over time in Black and Hispanic populations and high SVI zip codes further underscore the deeply embedded systemic nature of racialized disparities and the highly intersectional nature of systemic racism and social vulnerability [1,27–30,33–35]. Even though several vaccination strategies sought to prioritize Black and Hispanic individuals living in high SVI zip codes, given their high burden of disease earlier on [11,12,36–38], these groups still had dramatically lower vaccination rates compared to White and Asian individuals in the same high SVI zip codes and those from zip codes with low SVI. As the initial vaccine rollout progressed, though, vaccination rates in Black and Hispanic populations did eventually exceed those in White (though not Asian) populations. This coincided with wider vaccine availability and a shift toward vaccine administration at smaller centers such as pharmacies. Again, these changes in vaccination rates over time may be indicative of increased access to vaccinations in Black, Hispanic, and other socially vulnerable communities through community-based settings as opposed to large health systems [30,34,37,39]. These patterns must also be contextualized within the growing literature on vaccine confidence and hesitancy. Vaccine hesitancy is not monolithic and ranges from beliefs in conspiracy theories and skepticism about COVID-19 to more nuanced concerns regarding safety, side effects, inability to take time off work, observing others safely vaccinated (i.e., social proof), and lack of trusted messaging [29,33–35,40–42]; its patterns and trends across communities also vary [43,44]. Qualitative studies have shown that lack of vaccine confidence in Black communities in particular stems largely from histories of systematic mistreatment and racism—which include failed contemporary responses to COVID-19—leading to mistrust of larger institutions and concerns over bearing the burden of unfavorable safety and side effect profiles (particularly given the rapid timeline of vaccine development and shifting messaging over the need for additional doses) [29,35]. However, rates of primary series completion in the Black population also likely increased as confidence in vaccinations improved over time, more of the population was safely vaccinated (i.e., social proof), purposeful and targeted messaging was delivered from trusted sources, and there were more opportunities to discuss specific questions and concerns with trusted healthcare providers [43,44]. Although a common pattern with the diffusion of many innovations, it is critical to contextualize the structural disparities leading to this late adoption.

Although multiple strategies were put forth early in order prioritize equitable vaccination, our analysis shows that we were far from achieving such goals when examined from several metrics. Early vaccine allocation strategies designed to maximize benefits when supply was limited included considerations for prioritizing groups with higher risk for COVID-19 exposure or who had experienced higher burden of COVID-19 disease using metrics such as

**Table 3. Poisson model of individual- and zip code-level factors associated with receipt of primary COVID-19 vaccination series and booster.**

| Factor | Primary series | | | | Booster | | | |
|---|---|---|---|---|---|---|---|---|
| | Unadjusted risk ratio (95% CI) | p-Value | Adjusted risk ratio (95% CI) | p-Value | Unadjusted risk ratio (95% CI) | p-Value | Adjusted risk ratio (95% CI) | p-Value |
| **Race/ethnicity** | | | | | | | | |
| Black | 0.86 (0.86–0.86) | <0.001 | 0.94 (0.93–0.94) | <0.001 | 0.65 (0.66–0.66) | <0.001 | 0.83 (0.82–0.83) | <0.001 |
| White | 1 (REF) | | 1 (REF) | | 1 (REF) | | 1 (REF) | |
| Hispanic | 0.89 (0.88–0.89) | | 0.96 (0.95–0.97) | | 0.60 (0.59–0.60) | | 0.76 (0.75–0.77) | |
| Asian | 1.00 (0.99–1.00) | | 1.03 (1.02–1.03) | | 0.96 (0.95–0.97) | | 1.08 (1.07–1.09) | |
| Other | 1.72 (1.71–1.72) | | 1.65 (1.65–1.66) | | 1.88 (1.88–1.89) | | 1.76 (1.76–1.77) | |
| **Age category** | | | | | | | | |
| 12–19 years | 1.28 (1.28–1.29) | <0.001 | 1.27 (1.26–1.27) | <0.001 | 0.77 (0.77–0.78) | <0.001 | 0.76 (0.75–0.76) | <0.001 |
| 20–34 years | 0.83 (0.83–0.83) | | 0.84 (0.84–0.84) | | 0.59 (0.59–0.60) | | 0.61 (0.60–0.61) | |
| 35–44 years | 1.01 (1.01–1.02) | | 1.01 (1.01–1.01) | | 0.92 (0.92–0.93) | | 0.92 (0.92–0.93) | |
| 45–54 years | 1 (REF) | | 1 (REF) | | 1 (REF) | | 1 (REF) | |
| 55–64 years | 1.11 (1.11–1.12) | | 1.11 (1.10–1.11) | | 1.31 (1.30–1.32) | | 1.29 (1.28–1.30) | |
| 65–74 years | 1.34 (1.33–1.34) | | 1.30 (1.30–1.30) | | 1.93 (1.92–1.94) | | 1.83 (1.82–1.84) | |
| ≥75 years | 1.31 (1.31–1.31) | | 1.24 (1.24–1.25) | | 1.93 (1.92–1.94) | | 1.77 (1.76–1.78) | |
| **Sex** | | | | | | | | |
| Male | 1 (REF) | <0.001 | 1 (REF) | <0.001 | 1 (REF) | <0.001 | 1 (REF) | <0.001 |
| Female | 1.09 (1.08–1.09) | | 1.07 (1.07–1.07) | | 1.19 (1.18–1.19) | | 1.13 (1.13–1.14) | |
| **Zip-code-level characteristics** | | | | | | | | |
| Social Vulnerability Index | | | | | | | | |
| Low | 1 (REF) | <0.001 | 1 (REF) | <0.001 | 1 (REF) | <0.001 | 1 (REF) | <0.001 |
| Medium | 0.87 (0.87–0.87) | | 0.92 (0.91–0.92) | | 0.77 (0.77–0.77) | | 0.83 (0.82–0.83) | |
| High | 0.80 (0.80–0.81) | | 0.88 (0.88–0.89) | | 0.59 (0.58–0.59) | | 0.69 (0.68–0.69) | |
| Total population, per 10,000 increase | 1.04 (1.04–1.04) | <0.001 | 1.02 (1.02–1.02) | <0.001 | 1.06 (1.05–1.06) | <0.001 | 1.02 (1.02–1.02) | <0.001 |
| Percent Black, per 10% increase | 0.98 (0.98–0.98) | <0.001 | —* | —* | 0.95 (0.95–0.95) | <0.001 | —* | —* |
| Median income, per $15,000 increase | 1.05 (1.05–1.05) | <0.001 | —* | —* | 1.11 (1.11–1.11) | <0.001 | —* | —* |
| Percent below poverty line, per 2.5% increase | 0.97 (0.97–0.97) | <0.001 | —* | —* | 0.93 (0.93–0.93) | <0.001 | —* | —* |
| Percent without health insurance, per 2.5% increase | 0.96 (0.96–0.96) | <0.001 | —* | —* | 0.90 (0.90–0.90) | <0.001 | —* | —* |
| Percent in healthcare industry, per 2.5% increase | 1.02 (1.01–1.02) | <0.001 | —* | —* | 1.04 (1.04–1.04) | <0.001 | —* | —* |
| Percent in service industry, per 2.5% increase | 0.97 (0.97–0.97) | <0.001 | —* | —* | 0.92 (0.92–0.92) | <0.001 | —* | —* |
| Vaccine sites per 10,000, per 1 site increase | 1.01 (1.01–1.01) | <0.001 | 1.01 (1.01–1.01) | <0.001 | 1.01 (1.01–1.01) | <0.001 | 1.01 (1.01–1.01) | <0.001 |
| Cases per 100,000, per 1,500 increase | 1.01 (1.01–1.01) | <0.001 | 1.00 (1.00–1.00) | <0.001 | 1.01 (1.01–1.01) | | 0.99 (0.99–0.99) | <0.001 |
| Deaths per 100,000, per 50 increase | 1.00 (1.00–1.00) | <0.001 | 1.00 (1.00–1.00) | <0.001 | 1.00 (1.00–1.01) | | 1.00 (1.00–1.00) | 0.14 |
| **Region** | | | | | | | | |
| St. Louis | 1 (REF) | <0.001 | 1 (REF) | <0.001 | 1 (REF) | <0.001 | 1 (REF) | <0.001 |
| Kansas City | 0.92 (0.92–0.93) | | 0.95 (0.95–0.96) | | 0.87 (0.87–0.87) | | 0.93 (0.93–0.94) | |

Continuous variables are scaled so that a 1-unit increase represents approximately half of the interquartile range for that variable.

*Excluded from multivariable model due to collinearity with Social Vulnerability Index.

CI, confidence interval; REF, reference value.

geography, SVI, and race/ethnicity (in addition to using age, comorbidities, and high-risk occupations) [11–13,36–38]. Still, these strategies mostly focused on determining vaccine eligibility, but eligibility for or availability of vaccines doesn't equate to adequate access. Indeed, achieving equity would have also required early concomitant prioritization and efforts to target structural barriers to vaccine uptake and reasons for later adoption [45]. Several programs demonstrated success using early, low barrier, and widely available access to vaccines at community-based sites (as opposed to mass vaccination sites and large health systems, often requiring online registration) in areas with high social vulnerability, coupled with abundant opportunities to connect with and discuss concerns with trusted sources of information [30,34,41,46–50]. A program in San Francisco leveraged a community-based vaccination site near a transportation hub to target both access and trust-related barriers, and leveraged both high-touch (e.g., going door-to-door to provide information and register individuals) and low-touch methods (e.g., flyers and advertisements) [50]. Approaches like these are even more important during the later stages of vaccination rollout, when large or mass vaccination sites—which allow for high volume for those already eager to be vaccinated—are likely at the limits of their reach.

There are several limitations to our analysis. First, reporting of all vaccinations was mandated by the state, but race/ethnicity and zip code were not reported consistently, particularly at smaller sites. Still, as this missingness was highly dependent on vaccination date and site, multiple imputation would still yield unbiased results even with higher levels of missingness [24–26]. Second, there may also have been misclassification of zip codes of individuals if permanent addresses did not match where people were actually living at the time of vaccination, or in our categorization of vaccine location types. However, any misclassification was likely small, and there is no reason to believe that there was systemic error that would substantially bias our results. Third, we used zip code population estimates from the 2020 census data, but true population sizes—and thus the appropriate denominators for some analyses—may have changed since then, particularly due to the well-documented migrations that occurred during the early phases of the pandemic. Fourth, we lacked complementary data that could help contextualize our findings (e.g., association between race/ethnicity and time or location of vaccination) and help characterize the relationship with potential drivers of these disparities, such as data on occupation, health insurance status, linkage to primary care, and vaccination awareness, knowledge, beliefs, and intentions. Fourth, in this analysis we were unable to provide more granular details or include separate categories for other racial/ethnic minorities such as indigenous or multi-racial individuals, due to either small populations in the regions that would lead to unstable statistical estimates or the inability to link these population across data sources. Still, although we do include Black, White, Hispanic, and Asian communities, it remains critical to also assess disparities across other minoritized communities, acknowledging that the multidimensional nature of health disparities and unique drivers across these different communities warrant dedicated attention and public health action.

In conclusion, we provide nuanced characterizations of the disparities in COVID-19 vaccination across racial/ethnic communities, across levels of social vulnerability, over time, and across types of vaccine administration sites after 1 year of vaccination. Equitable COVID-19 vaccination is one of the most critical targets for successfully ending the pandemic, but, despite substantial discussion on how to effectively do so, it is clear that our strategies—both nationally and in Missouri—have yet to overcome the deeply entrenched systemic inequities in healthcare and society. Future planning for proactive and considered public health strategies in the face of pandemic emergencies—as opposed to reactive approaches—is needed to ensure that our responses are equitable from the outset and do not disproportionately affect minority communities both in the United States and globally.

## Supporting information

**S1 Fig. Cumulative incidence of primary COVID-19 vaccination series and booster completion by race/ethnicity and SVI.**
(DOCX)

**S2 Fig. St. Louis: Rates and cumulative incidence of receiving the primary COVID-19 vaccination series and boosters by race/ethnicity and SVI over time.**
(DOCX)

**S3 Fig. Kansas City: Rates and cumulative incidence of receiving the primary COVID-19 vaccination series and boosters by race/ethnicity and SVI over time.**
(DOCX)

**S4 Fig. Rates of diagnosed cases and deaths from COVID-19.**
(DOCX)

**S5 Fig. Lorenz curves of disparities in COVID-19 vaccinations over time—primary series.**
(DOCX)

**S6 Fig. Lorenz curves of disparities in COVID-19 vaccinations over time—booster.**
(DOCX)

**S7 Fig. Disparities in COVID-19 primary vaccine series and boosters among Black, Hispanic, and Asian versus White residents in the same zip code over time.**
(DOCX)

**S1 STROBE Checklist. STROBE checklist.**
(DOCX)

**S1 Table. Characteristics of individuals completing the primary vaccine series.**
(DOCX)

**S2 Table. Characteristics of individuals receiving a booster vaccination.**
(DOCX)

**S3 Table. Rates of initiating/completing primary vaccine series and boosters by race/ethnicity and SVI.**
(DOCX)

**S4 Table. Characteristics of zip codes by quartile of Lorenz curve—number of COVID-19 vaccinations relative to total population.**
(DOCX)

**S5 Table. Characteristics of zip codes by quartile of Lorenz curve—number of COVID-19 vaccinations relative to diagnosed COVID-19 cases.**
(DOCX)

**S6 Table. Characteristics of zip codes by quartile of Lorenz curve—number of COVID-19 vaccinations relative to deaths due to COVID-19.**
(DOCX)

**S7 Table. Characteristics of zip codes by quartile of Lorenz curve—number of COVID-19 vaccinations relative to total SVI.**
(DOCX)

## Author Contributions

**Conceptualization:** Aaloke Mody, Cory Bradley, Matifadza G. Hlatshwayo, Vetta Sanders-Thompson, William G. Powderly, Elvin H. Geng.

**Data curation:** Aaloke Mody, Branson Fox, Anne Trolard, Khai Hoan Tram.

**Formal analysis:** Aaloke Mody, Salil Redkar.

**Investigation:** Aaloke Mody, Branson Fox, Anne Trolard, Franda Thomas, Matt Haslam, George Turabelidze, Elvin H. Geng.

**Methodology:** Aaloke Mody, Ingrid Eshun-Wilson, Lindsey M. Filiatreau, Elvin H. Geng.

**Project administration:** Anne Trolard.

**Resources:** Anne Trolard, George Turabelidze, Elvin H. Geng.

**Validation:** Aaloke Mody.

**Writing – original draft:** Aaloke Mody.

**Writing – review & editing:** Aaloke Mody, Cory Bradley, Salil Redkar, Branson Fox, Ingrid Eshun-Wilson, Matifadza G. Hlatshwayo, Anne Trolard, Khai Hoan Tram, Lindsey M. Filiatreau, Franda Thomas, Matt Haslam, George Turabelidze, Vetta Sanders-Thompson, William G. Powderly, Elvin H. Geng.

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
