## [Editor Report · Decision Letter 0]

8 Apr 2022

Dear Dr Mody, 

Thank you for submitting your manuscript entitled "Characterizing Equity in COVID-19 Vaccine Distribution: A Population-Level Analysis quantifying disparities across social vulnerability, race, location, and time." for consideration by PLOS Medicine.

Your manuscript has now been evaluated by the PLOS Medicine editorial staff and I am writing to let you know that we would like to send your submission out for external assessment.

However, we first need you to complete your submission by providing the metadata that is required for full assessment. To this end, please login to Editorial Manager where you will find the paper in the 'Submissions Needing Revisions' folder on your homepage. Please click 'Revise Submission' from the Action Links and complete all additional questions in the submission questionnaire.

Please re-submit your manuscript within two working days, i.e. by Apr 12 2022 11:59PM.

Once your full submission is complete, your paper will undergo a series of checks in preparation for assessment. 

Kind regards,

Richard Turner, PhD

rturner@plos.org

---

## [Decision Letter · Decision Letter 1]

18 May 2022

Dear Dr. Mody,

Thank you very much for submitting your manuscript "Characterizing Equity in COVID-19 Vaccine Distribution: A Population-Level Analysis quantifying disparities across social vulnerability, race, location, and time." (PMEDICINE-D-22-01093R1) for consideration at PLOS Medicine. 

Your paper was discussed with an academic editor with relevant expertise and sent to independent reviewers, including a statistical reviewer. The reviews are appended at the bottom of this email and any accompanying reviewer attachments can be seen via the link below:

[LINK]

In light of these reviews, we will not be able to accept the manuscript for publication in the journal in its current form, but we would like to invite you to submit a revised version that addresses the reviewers' and editors' comments fully. You will recognize that we cannot make a decision about publication until we have seen the revised manuscript and your response, and we expect to seek re-review by one or more of the reviewers. 

We hope to receive your revised manuscript by Jun 13 2022 11:59PM. Please email us (plosmedicine@plos.org) if you have any questions or concerns.

Please let me know if you have any questions, and we look forward to receiving your revised manuscript. 

Sincerely,

Richard Turner, PhD

Senior editor, PLOS Medicine

rturner@plos.org

Please adapt the competing interest statement (submission form) to include "EHG is a member of PLOS Medicine's Editorial Board" or similar. 

We ask you to restructure the title to better match journal style and suggest: "Quantifying disparities in COVID-19 vaccine distribution by social vulnerability, ethnicity, location, and time: A population-level analysis".

Please also state where the study was done in the title. 

Please add a new final sentence to the "Methods and findings" subsection of your abstract, which should begin "Study limitations include ..." or similar and should quote 2-3 of the study's main limitations. 

After the abstract, please add a new and accessible "Author summary" section in non-identical prose. You may find it helpful to consult one or two recent research papers published in PLOS Medicine to get a sense of the preferred style. 

The sentence at line 99 beginning "The novel application ..." is really an item of discussion, and so should be removed or relocated to the Discussion. 

Early in the Methods section (main text), please state whether the study had a protocol or prespecified analysis plan, and if so attach the relevant document as a supplementary file, referred to in the text. 

Please highlight any non-prespecified analyses. 

We suggest substituting "ethnicity" for "race" throughout. 

Please remove the information on funding and competing interests from the end of the main text. In the event of publication, this information will appear in the article metadata via entries in the submission form, and does not need to be duplicated. 

In the reference list, please use the journal name abbreviation "PLoS ONE".

Noting reference 4 and others, please ensure that all references have full access details. 

Please add a completed checklist for the most appropriate reporting guideline, e.g., STROBE, as an attachment, labelled "S1_STROBE_Checklist" or similar and referred to as such in the Methods section (main text). 

In the checklist, please ensure that individual items are referred to by section (e.g., "Methods") and paragraph number, not by line or page numbers as these generally change in the event of publication. 

Comments from the reviewers:

*** Reviewer #1: 

This is an excellent, well-written article that brings a health equity lens to the rollout of SARS-CoV-2 vaccination in two large cities in Missouri. The data are population-based and the study produced collaboratively by academic researchers and health department staff. Despite COVID-19 vaccines being freely available, how they are made available will affect who gets them quickly enough to derive the most reduced risk in the burden of infection (hospitalizations, long COVID, deaths). The authors contextualize their findings in terms of both historical systemic racism in medicine and public health, as well as how implementation intended to counter inequities did not prevent what ended up being substantial racial/ethnic and social inequity in the uptake of vaccines. I have a few comments and suggestions intended to help improve the quality and impact of their manuscript:

1. Is there a way to examine, or at least discuss, vaccine uptake as it relates to timing of major surges. That is, in addition to having lower cumulative vaccination rates, were minoritized and vulnerable communities also less likely to be vaccinated in advance of the delta and omicron surges? This speaks to the timing of vaccination, which is an important dimension. Some of it is alluded to in Figure 1.

2. Outcome of vaccine uptake. There has been some concern about the quality of vaccine coverage data as derived from registries (e.g., double counting, denominator issues). For example, vaccine coverage rates well above 100% for some racial/ethnic groups. Related, 2018 census data may not reflect differential in and out migration that occurred in many jurisdictions during the early phase of the pandemic especially. The authors should at least acknowledged this as a potential limitation. But if they did any investigative work around this with their own data, it would be worth highlighting so that other jurisdictions factor these issues in as well.

3. The authors describe how much of the initial vaccine rollout was through large medical centers and health care providers. They note that pre-existing disparities in health care access barriers would be perpetuated under this model, which is true. But I think it is worth pulling back the lens even more. Getting vaccinated is not health care. It is a preventive action that requires broad reach in communities to achieve the requisite coverage. By definition, focusing on medical centers and the health care system as points of distribution for an intervention that requires broad community reach outside the health care system is in my view one key example of how and why disparities were created early on. The health care system, hospitals, and medical providers have no public health mandate and no management/implementation expertise in this areas. Thus, they are always going to be the wrong choice for the major node of implementation of a large scale public health initiative that by definition must reach those outside the health care system. Suggest a re-framing.

4. I am having a heard time wrapping my head around how a sample sizes of 32K Hispanic, 38K Asian, and 187K multiracial are too small to produce useful estimates (Discussion, Line 361), especially since they are not samples, but near full enumerations of those populations in each city. Please consider presenting more nuanced groupings than Black-White-Other.

5. For the SVI variable in Table 1, consider presenting low, medium, and high so readers can appreciate any gradient that may be there. Related, did the authors check that relationships were in fact linear in models where continuous exposures were examined (i.e., in models presented in Table 2)? If so, this should be stated in the methods. If not, the authors should examine these variables with fewer smoothing assumptions (e.g., as 3+ level categorical variables).

*** Reviewer #2: 

I mostly confine my remarks to statistical aspects of this paper. The overall method is sound, but I have a few issues to resolve before I can recommend publication

I will note that it could use some more editing for English usage. There are doubled words (had had), misplaced articles ("endured the disproportionate" should not have a "the"), wrong words ("empiric" should be "empirical") and so on. I don't usually comment on grammar, but here, it affects readability and may affect credibility. 

One non-statistical comment: I think the title should reflect that this study was only in two cities in one state in the USA. As is, it sounds universal.

Still on the title, but now statistical: I think it should be "over time" not "and time" -- disparities across social vulnerability, race, and location, over time. After all, while it is true that you can be inequitable regarding place, race, etc, it's hard to be inequitable about time -- July isn't going to complain that it got fewer vaccines than April. Another possibility to end with "controlling for time". 

p. 5 line 134 What are "health facilities"? Are these hospitals? And you aren't really looking at the size of the facility, but at its vaccination rate. This will surely be related to size, but also to other factors, such as the existence of other facilities nearby. (If a big hospital is near a mass vaccination site, people may go to the latter). 

p. 6 line 154-156 I wouldn't do this, but, since you also treated this as a continuous variable, it's not so bad.

 line 160 Rather than % of the population, this should be based on a number of people. ZIP codes are very disparate in population. I know that, in New York City, some ZIP codes have no residents at all (these are large office buildings that have their own ZIP). In Missouri, ZIP codes can have as few as 6 people (63464) and as many as 75,000 (63376) (See https://www.missouri-demographics.com/zip_codes_by_population). The latter one is actually in your data. Just checking St Louis (because I got interested) 63140 has only 347 people). 

Despite all the above, I think this is a good paper that uses somewhat unusual statistical methods in a good way. 

Peter Flom

*** Reviewer #3: 

Summary

Differences in vaccine coverage between populations may result in large disparities in COVID-19 morbidity and mortality. In this paper, the authors assess inequities in COVID-19 vaccine coverage in 7 counties in St. Louis and 4 counties in Kansas City throughout the first year of vaccine availability. The authors specifically look at Black and White populations in these counties and compare vaccination rates relative to the following measures: time, race, zip-code level social vulnerability index (SVI), vaccine location type, and COVID-19 disease burden. The authors summarize inequality using Lorenz curves and Gini coefficients, tools/methods from economics. The key findings from this study are that large inequities in vaccination between Black and White communities were seen early on during vaccine distribution and that inequities only decreased once methods of distribution broadened, and that inequities persist. This is significant because it highlights the importance of public health strategies taking into account structural drivers of inequities. 

Strengths

The paper is very well written. The authors do a great job outlining why research on inequities in vaccination rates should look at multiple structural and social determinants to gain a better understanding of how such inequities may be addressed. The methods are well justified and the use of traditional economics methods to describe health inequities is creative. The methods are described clearly in an accessible way that made it so even readers who are not well versed in statistics could understand at a basic level what was done. The graphs and figures are good, but could be made even more clear (details below). 

Critiques

1. The paper is an excellent retrospective look at vaccine rollout. However, I expected a little bit more about what cities or counties can do moving forward in the Omicron era. E.g., Figure 1 looks like cumulative proportion of 1 vaccine dose, would it be fairly easy for authors to look at 2 doses and booster? Given where discussion is now with fully vaccinated meaning at least 2 doses and booster availability, do these results hold up? A full look at booster coverage may be out of scope of the paper. However, there is concern about people not getting boosters and people (disproportionately Black, high SVI) who got their vaccine later were not eligible for boosters during the early Omicron wave. Some additional analysis - or discussion - of this point would be helpful to guide policy makers in applying the findings to the current moment. 

2. I have several suggestions to make the tables and figures even clearer:

- Across all tables and figures, there should be more clarity around what receipt of the COVID-19 vaccination is. Is it 1 dose or 2? Does it include boosters? Does it mean something different at different age groups? This should be clear in the title or note to each table/figure.

- Table 2. The regression coefficients for zip-code characteristics are difficult to compare with each other. I'd suggest standardizing the regression coefficients (dividing by the standard deviation of the predictor) to enable comparisons. 

- Fig 1C and 1D yscale should go 0 to 100% if everyone in the denominator is eligible.

- Figure 1, the lines all have colors in a similar tone (making it difficult for the colorblind) and making high/low SVI and White/Black comparisons difficult. I recommend changing the colors/patterns to make the graph easier to read. E.g., change lines to two colors, one color for low SVI and another for high SVI, and then solid lines for Black and dashed lines for White. (or something similar) That change would make it much easier to read this graph.

- Figure 2. Font is too small. Consider month abbreviations, consider removing the decimals on the percentages… should be easier to read

- Figure 3 - Color coding is interesting and could help people understand intuitively what the Lorenz curves are doing. Where blues and reds are more mixed, the line is closer to diagonal (less inequality) where blues are together and reds are together is where curves are far away from diagonal (more inequality). However, it is rather hard to see colors. Is there a way to make the colors (dots) bigger?

- Both of the last two graphs are titled "Figure 4". 

- The first "Figure 4". Gini coef is a measure of inequality (not disparity), hence the title should refer to trends in inequality, as measured by Gini coef.

- The first "Figure 4". I find the 4 lines to be distracting. My eye wanted to compare the lines with each other and understand the difference. But the MAIN point of the figure is that the lines fall over time (as indicated in the title of the figure) and that inequity remains high at the end. If the focus is on temporal trends, I think the figure could be improved by showing just the SVI line - and perhaps showing STL and KC on the same plot.

- The second "Figure 4" (i.e. Figure 5 - with 8 panels). I think there's too much going on here. I'd encourage the authors to simplify the figure so the main point is clearer. E.g., panels A-D are enough. Panel A/B show that SVI was correlated with vaccine uptake in blacks, but not in whites (HUGELY IMPORTANT FINDING); panel C/D show that blacks are substantially undervaccinated relative to whites when accounting for case rates. Panels E/F - the data are a bit all over the place, maybe a small numbrs issue with zip-level death rates. Panels G/H - I don't understand what "total SVI in black/white population" means. If the point is that vaccine coverage in inversely associated with social vulnerability, then that point has already been made earlier in the paper.

3. STL vs. KC. There were some notable differences between St Louis and Kansas City counties, but little discussion of these differences. Did the cities take any different approaches that affected vaccine coverage?

In all, the authors have conducted a timely and important analysis of small area vaccination inequities. Their emphasis on the role of concrete interventions (decentralized vaccine delivery) to mitigate vaccine inequities is a major strength of the paper. The findings will be valuable for policy makers, in addition to pushing the literature forward.

Jacob Bor

***

[LINK]

---

## [Decision Letter · Decision Letter 2]

26 Jul 2022

Dear Dr. Mody,

Thank you very much for re-submitting your manuscript "Quantifying Inequities in COVID-19 Vaccine Distribution Over Time by social vulnerability, race and ethnicity, and location: A Population-Level Analysis in St. Louis and Kansas City, Missouri" (PMEDICINE-D-22-01093R2) for review by PLOS Medicine.

I have discussed the paper with my colleagues and it was also seen again by three reviewers. I am pleased to say that provided the remaining editorial and production issues are dealt with we are planning to accept the paper for publication in the journal.

[LINK]

We look forward to receiving the revised manuscript by Aug 02 2022 11:59PM.   

Sincerely,

Beryne Odeny, PhD

PLOS Medicine

plosmedicine.org

Requests from Editors:

Comments from Reviewers:

Reviewer #1: The authors have been very responsive to reviewer and editorial comments.

Reviewer #2: The authors have addressed my concerns and I now recommend publication

Reviewer #3: The authors have responded to all of my prior concerns in this revision. The manuscript has been strengthened in particular by the full data update (no small amount of work) to include analyses of the full vaccination series and booster coverage. The expanded focus to look at other racial/ethnic groups is welcome, and the decision to de-emphasize the KC vs. STL distinction is reasonable based on the large amount of content presented. The exposition and exhibits are clearer as a result of the suggestions of other reviewers and Editors. The definitions of equity, race/ethnicity, and minoritized are careful and accurate to the U.S. context. The authors should mention explicitly that there were not enough American Indian / Alaska Native persons in the population to be included as a separate category with statistically stable estimates. (There is some understandable consternation in the AIAN research community about being lumped into "Other" without justification.) I have no further critiques of this fine paper.

[LINK]

---

## [Editor Report · Decision Letter 3]

2 Aug 2022

Dear Dr Mody, 

On behalf of my colleagues and the Academic Editor, Dr. Nicola Low, I am pleased to inform you that we have agreed to publish your manuscript "Quantifying Inequities in COVID-19 Vaccine Distribution Over Time by social vulnerability, race and ethnicity, and location: A Population-Level Analysis in St. Louis and Kansas City, Missouri" (PMEDICINE-D-22-01093R3) in PLOS Medicine.

PRESS

Sincerely, 

Beryne Odeny 

PLOS Medicine